# Direct observation of intrinsic twin domains in tetragonal $CH_3NH_3PbI_3$

Mathias Uller Rothmann[1,*], Wei Li[1,*], Ye Zhu[1,*], Udo Bach[1,2,3], Leone Spiccia[4], Joanne Etheridge[1,5] & Yi-Bing Cheng[1,6]

Organic–inorganic hybrid perovskites are exciting candidates for next-generation solar cells, with $CH_3NH_3PbI_3$ being one of the most widely studied. While there have been intense efforts to fabricate and optimize photovoltaic devices using $CH_3NH_3PbI_3$, critical questions remain regarding the crystal structure that governs its unique properties of the hybrid perovskite material. Here we report unambiguous evidence for crystallographic twin domains in tetragonal $CH_3NH_3PbI_3$, observed using low-dose transmission electron microscopy and selected area electron diffraction. The domains are around 100–300 nm wide, which disappear/reappear above/below the tetragonal-to-cubic phase transition temperature (approximate 57 °C) in a reversible process that often 'memorizes' the scale and orientation of the domains. Since these domains exist within the operational temperature range of solar cells, and have dimensions comparable to the thickness of typical $CH_3NH_3PbI_3$ films in the solar cells, understanding the twin geometry and orientation is essential for further improving perovskite solar cells.

[1] Department of Materials Science and Engineering, Monash University, Clayton, Victoria 3800, Australia. [2] Commonwealth Scientific and Industrial Research Organization, Manufacturing Flagship, Clayton, Victoria 3168, Australia. [3] Melbourne Centre for Nano Fabrication, 151 Wellington Road, Clayton, Victoria 3168, Australia. [4] School of Chemistry, Monash University, Clayton, Victoria 3800, Australia. [5] Monash Centre for Electron Microscopy, Monash University, Clayton, Victoria 3800, Australia. [6] State Key Laboratory of Advanced Technology for Materials Synthesis and Processing, Wuhan University of Technology, Wuhan 430070, China. * These authors contributed equally to this work. Correspondence and requests for materials should be addressed to J.E. (email: joanne.etheridge@monash.edu) or to Y.-B.C. (email: yibing.cheng@monash.edu).

Organic–inorganic hybrid perovskites of the type $ABX_3$ (A = organic cation; B = Ge, Sn, Pb and X = halogen) have achieved astonishing breakthroughs in the field of photovoltaics and optoelectronics. The power conversion efficiencies of perovskite solar cells (PSC) have increased rapidly from an initial 3.8% in 2009 (ref. 1) to a recent 22.1% (ref. 2). In spite of the numerous papers published on the application of these materials in solar cells, an in-depth understanding of the crystal structure and microstructure, and their influence on the physical properties of the hybrid perovskite is still lacking.

$CH_3NH_3PbI_3$ is the most widely studied organic–inorganic hybrid perovskite. It has been reported that $CH_3NH_3PbI_3$ undergoes transitions from cubic to tetragonal at $\sim 330\,K$ (as temperature is decreased) and then from tetragonal to orthorhombic at $\sim 165\,K$ (refs 3,4). However, the unambiguous determination of the space group of $CH_3NH_3PbI_3$ has proved challenging due to structural complexities, such as disorder in both the organic and inorganic components[5,6] and possibly twinning[7,8]. In particular, for the practically important, room temperature tetragonal phase, two possible space groups have been proposed: the centrosymmetric, hence non-polar, space group I4/mcm (refs 3,5,6,8,9) or the non-centrosymmetric, polar space group I4cm (ref. 7). This is an important question to resolve. Crystal structure controls properties, including ferroelectricity, which has been proposed to possibly play a role in the photovoltaic properties of $CH_3NH_3PbI_3$ (refs 10,11). For example, spontaneous polarization, or ferroelectricity, has been suggested to be responsible for the efficient separation of photoexcited electron–hole pairs, which might explain the superior performance of $CH_3NH_3PbI_3$ in solar cells[12]. Liu et al.[11] proposed that ferroelectric domains are a factor responsible for the hysteresis in current density-voltage curves of $CH_3NH_3PbI_3$-based solar cells. However, others have argued that $CH_3NH_3PbI_3$ is not a ferroelectric material due to lack of credible evidence in property measurements[13]. Experimental efforts to detect ferroelectricity directly in $CH_3NH_3PbI_3$ have so far yielded mixed results[13–17].

One of the structural complexities that can hinder determination of the space group, and hence the atomic structure, is twinning. However, the evidence for the presence and nature of twinning in $CH_3NH_3PbI_3$ is not yet clear. Fang et al.[8] and Stoumpos et al.[7] both incorporated pseudo-merohedral twinning into refinements of X-ray diffraction data from nominally single-crystal tetragonal $CH_3NH_3PbI_3$ but with different outcomes; Fang et al.[8] obtained a much better fit to the non-polar I4/mcm space group than to I4cm, whereas Stoumpos et al.[7] found the opposite. Recently, Hermes et al.[17] observed nanoscale-striped domains in the electromechanical response of a polycrystalline thin film of tetragonal $CH_3NH_3PbI_3$ using piezoresponse force microscopy (PFM). They proposed that the stripes were due to ferroelastic twin domains with a polarization oriented in the $a_1$-$a_2$-phase with a 45° angle to the $\{110\}_t$ surface (throughout this manuscript, the subscripts 't' and 'c' denote indexing in the tetragonal phase and cubic phase, respectively). Given the importance of possible twin domains for understanding the intrinsic atomic structure and properties of this material, as well as its application in photovoltaic devices, there is a need to obtain unequivocal evidence for twinning in $CH_3NH_3PbI_3$ and to determine its scale and geometry.

Here we report direct imaging and diffraction analysis of twin domains in $CH_3NH_3PbI_3$ using transmission electron microscopy (TEM). TEM is a classic method used to detect twin domains and determine their geometry[18,19]. However, to the best of our knowledge, twin domains have not yet been reported in TEM studies of $CH_3NH_3PbI_3$. We suspect that this is due to the extreme sensitivity to electron irradiation of $CH_3NH_3PbI_3$

(ref. 20). With this in mind, this study was undertaken using specialized and carefully established low-dose (around $1\,e\,\text{Å}^{-2}\,s^{-1}$), rapid acquisition conditions. Using this approach, we have been able to determine the size, morphology and crystallography of the twin domains in $CH_3NH_3PbI_3$. Importantly, the size of the twin domains (around 100–300 nm) is comparable to the thickness of the perovskite layer (around 300 nm) in a photovoltaic device. Given that the twin domains are observed to exist well within the operative temperature range of a solar cell, our observations have opened up a dimension for investigation of the effects of the crystal structure and microstructure on the performance of PSC.

## Results

**Direct observation of crystallographic twinning.** A typical bright-field 200 kV TEM image of a room temperature $CH_3NH_3PbI_3$ thin film is shown in Fig. 1. Most of the grains exhibit a 'stripe contrast', that is, parallel bands of alternate bright and dark contrast, each band being $\sim 100$–300 nm wide (as highlighted by the blue circles in Fig. 1a). When examining the surface of the film using a scanning electron microscope (SEM), we did not observe any morphology consistent with the stripe contrast, as seen in Fig. 2. This strongly suggests that surface morphology is not the origin of this periodic stripe structure. Selected area electron diffraction (SAED) patterns taken from individual $CH_3NH_3PbI_3$ grains near the $<110>$ zone axis clearly show the 'split spots' characteristic to twin domains (Fig. 1b). Indexing the SAED pattern from a given grain shows that it comprises two overlapping single-crystal diffraction patterns of tetragonal $CH_3NH_3PbI_3$ with a mirrored orientation relationship, as occurs for twin domains[18,19]. The diffraction spots from adjacent domains are mirrored with respect to each other across the twin axis (perpendicular to the twin plane). This results in a row of common diffraction spots from each domain, which are coincident along the twin axis, with all other spots from the two domains very slightly separated and the separation increasing with distance from the twin axis. This separation of twin reflections is a classic signature of twinning[18,19] and is often described informally as 'split spots', although the two spots are separate reflections, deriving from adjacent domains. This is clearly identified in the enlarged region of Fig. 1b shown in Fig. 1c. A comprehensive survey of reciprocal space was undertaken (see TEM characterization) and all diffraction patterns were consistent with this twinning geometry. No other intrinsic twin geometries were observed.

The different crystallographic orientations of adjacent twin domains can lead to different sets of electron beam reflections from these adjacent domains being transmitted via the objective aperture to form the TEM image. This in turn leads to adjacent domains having a different image intensity, thus generating the stripe contrast visible in Fig. 1a. Hence, we conclude that the observed striped pattern in Fig. 1a reflects twin domains in tetragonal $CH_3NH_3PbI_3$. It is noted that some of the grains in Fig. 1a will be oriented so that the electron beam is at a significant angle to the twin boundary. In these projections, little contrast will be visible. In particular, if the angle is 90°, no contrast will be visible. This twinning is a bulk phenomenon, since there is no evidence for an untwinned phase in the diffraction patterns of the twinned grains.

**Crystallographic model for twinning.** The diffraction pattern in Fig. 1b allows the geometry of the $CH_3NH_3PbI_3$ twin domains to be derived, as illustrated in Fig. 3. The separation of the diffraction spots from the two crystallographic twin orientations

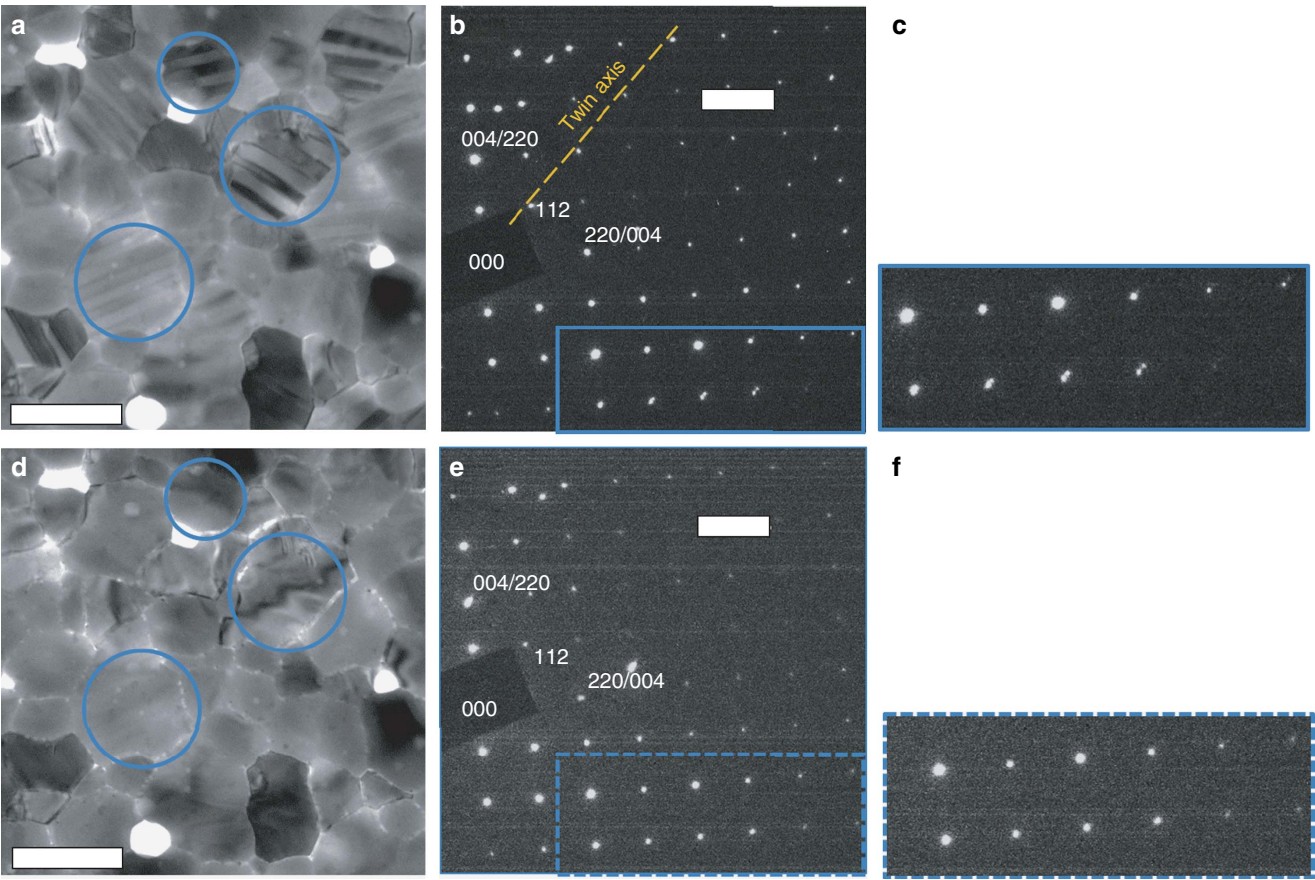

**Figure 1 | TEM images and SAED patterns of CH₃NH₃PbI₃ thin film at room temperature.** (**a**) Bright-field TEM image of pristine $CH_3NH_3PbI_3$ thin film at room temperature. A stripe contrast is visible through some of the grains (examples circled in blue). (**b**) Near $[1\bar{1}0]_t$-oriented diffraction pattern taken from a grain exhibiting stripe contrast showing two single-crystal patterns with a mirrored relationship. Coincident hh2h spots lie along the twin axis with all other spots from the two domains very slightly separated and the separation increasing away from the twin axis, as seen in the magnified region (solid blue rectangle) in **c**. (**d**–**f**) The same region as that in **a**,**b** but after extended electron beam exposure at a dose rate of around $1 e Å^{-2} s^{-1}$. The stripe contrast and 'split' spots are gone. All indexes are in the tetragonal phase. The scale bars in **a**,**d** are 500 nm, and the ones in **b**,**e** are $2 nm^{-1}$.

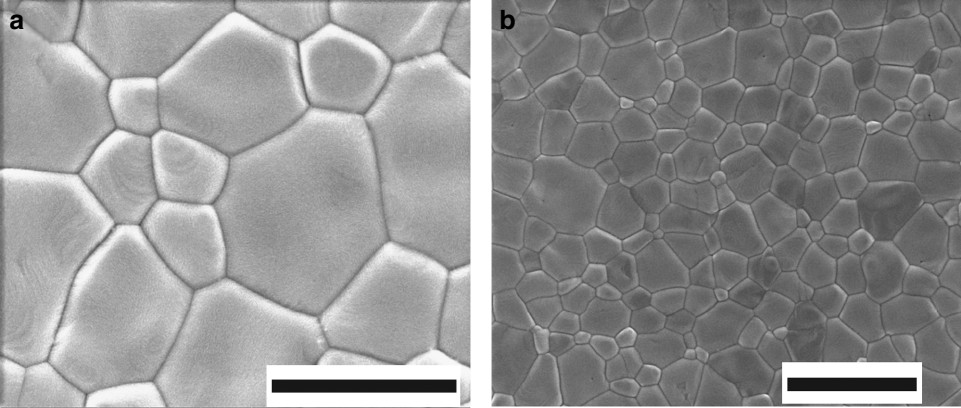

**Figure 2 | SEM image of the surface morphology of a CH₃NH₃PbI₃ thin film.** The $CH_3NH_3PbI_3$ film is spin-coated on ultra-thin carbon-coated copper TEM grid, using the same preparation method as the films in Figs 1, 4, 5 and 6. No morphology resembling the contrast stripes is observed. Scale bar, 1 μm (**a**); 2 μm (**b**).

reduces to zero along the dashed line (Fig. 3b), corresponding to the twin axis[21]. This line passes through hh2h reflections, indicating that the mirror plane of the twinning structure is parallel to $\{112\}_t$. (SAED patterns and images correlated from the same region are given in Fig. 4). In the room temperature tetragonal structure of $CH_3NH_3PbI_3$, there is a slight difference in the lattice spacing of the $\{110\}_t$ and $\{002\}_t$ planes ($d_{002} > d_{110}$)[22]. This difference gives rise to a small deviation angle (θ) from

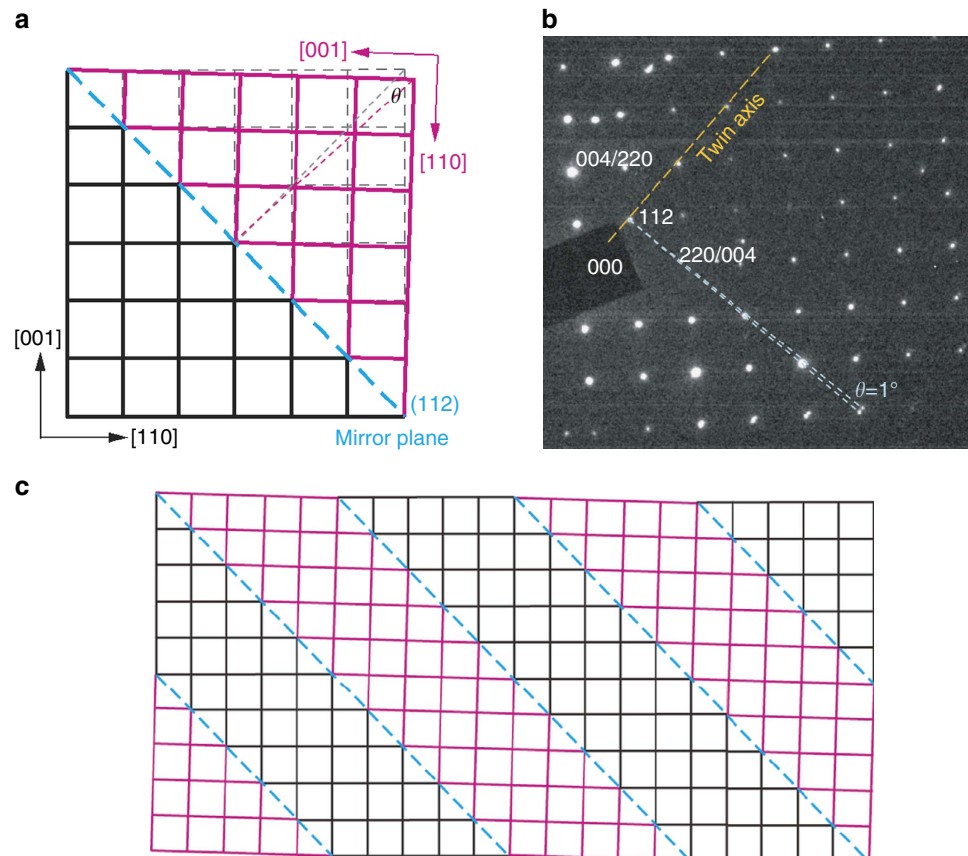

**Figure 3 | Schematic of proposed twinning structure in CH₃NH₃PbI₃.** (**a**) Schematic of the proposed twinning geometry in $[1\bar{1}0]_t$-oriented $CH_3NH_3PbI_3$ lattice. The original lattice without twinning is drawn in dashed thin lines. (**b**) The same SAED pattern as Fig. 1b. (**c**) Schematic of the proposed twin-domain structure in $[1\bar{1}0]_t$-oriented $CH_3NH_3PbI_3$ lattice. All indexes are in the tetragonal phase.

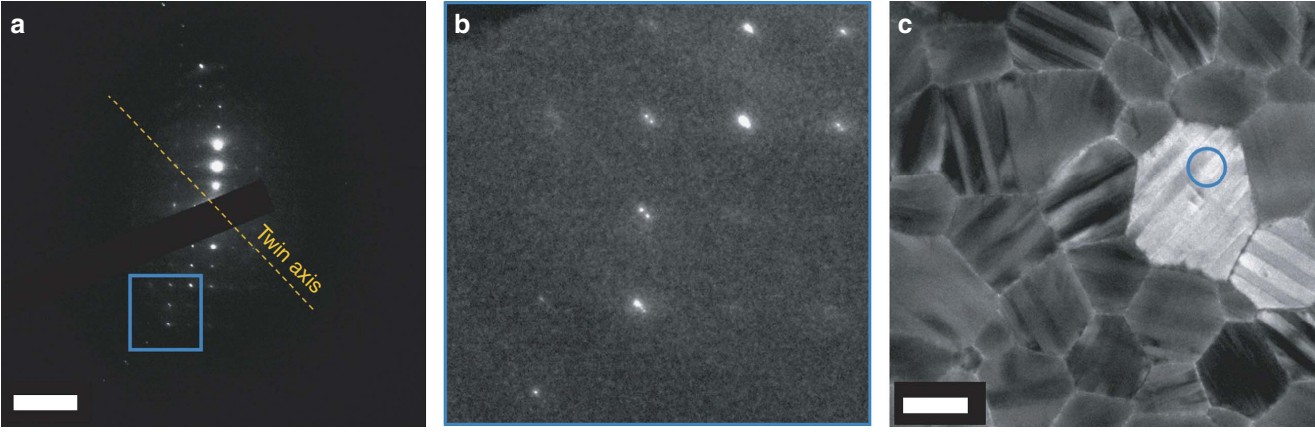

**Figure 4 | SAED pattern of the CH₃NH₃PbI₃ grain with striped contrast.** (**a**) Diffraction pattern near $[1\bar{1}0]_t$ zone axis obtained from the area in the blue circle in **c**. Double spots are visible in **b**, which is the highlighted regions in **a**. (**c**) Grain from which the diffraction pattern in **a** was obtained. Scale bar, 5 nm⁻¹ (**a**); 500 nm (**b**).

90° between the lattices mirrored across $\{112\}_t$ (Fig. 3a), which further leads to the separation of the mirrored diffraction spots in Fig. 3b. The separation angle can be estimated from $d_{110}/d_{002}$ (around 0.99)[22] with the formula:

$$\theta = \pi - 4\tan^{-1}(d_{110}/d_{002})$$

The derived angle is about 1° (or equivalently, there is around 89° between the $<001>_t$ directions in adjacent twin domains),

which is in excellent agreement with the measured angle of separation in Fig. 3b, validating our twinning model in Fig. 3a. Quarti's theoretical analysis also reported that at room temperature, experimental X-ray diffraction patterns of $CH_3NH_3PbI_3$ match better with a mixed structure that contains tilting of one of the octahedra around $[001]_t$ and the other tilting around $[110]_t$ (ref. 12). Such a mixed-tilting model is actually equivalent to having domains of octahedra tilt around

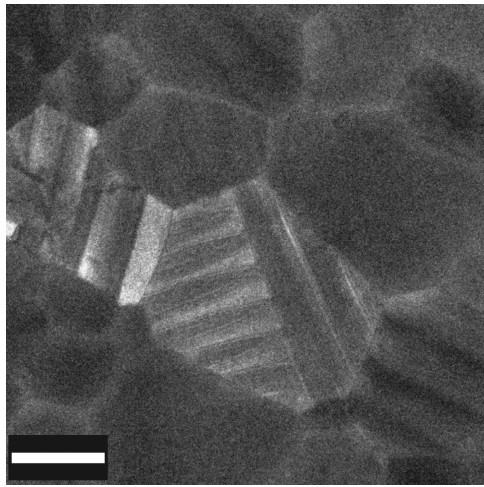

**Figure 5 | The CH₃NH₃PbI₃ grain with needle-like twin-domain boundaries.** Grain from a CH₃NH₃PbI₃ thin film showing two sets of twin domains orthogonal to each other representing different members of the symmetry-equivalent family of planes {112}ₜ (for example, (112)ₜ and (11$\bar{2}$)ₜ, which are oriented around 89.5° to each other). (To avoid the additional electron dose that would be incurred through tilting, the grain was imaged 'as-found', tilted off the <110>ₜ zone axis.) The 'horizontal' domains taper as they approach the intersection with their orthogonal analogue: this is behaviour typical of twin domains in a wide range of ferroelastic crystals[38]. These needle-like shapes form to minimize strain arising from the interaction with an orthogonal domain (or other local 'defect')[39]. The resultant stress provides sufficient energy to enable a local deviation from the primary twin orientation[23]. Scale bar, 500 nm.

[001]ₜ, which swap direction with [110]ₜ across neighbouring domains as demonstrated in the twinning model in Fig. 3.

The twin contrast observed in our TEM images, including orthogonal domain boundaries and associated needle shapes typical of ferroelastic crystals[23] (see Fig. 5), is remarkably similar in size to the recent PFM observation of a striped contrast in <110>ₜ-oriented CH₃NH₃PbI₃ by Hermes et al.[17], which these authors suggested is due to ferroelastic twin domains. From this, it appears that the twinning structure observed here might be intrinsic to tetragonal CH₃NH₃PbI₃ and not a manifestation of our specific synthesis. However, the combination of our TEM imaging and SAED provides direct evidence for a twinning structure model (Fig. 3a) that is different from the $a_1$-$a_2$-phase model deduced from the PFM results[17]. The observed twin structure is similar to the classic 90° $a$-$a$ domains present in inorganic perovskite oxides such as tetragonal BaTiO₃ (ref. 24). The inability of PFM to measure the crystallographic orientation of the twin-domain boundaries might partly contribute to the discrepancy between the $a_1$-$a_2$-phase model and our direct measurement.

In summary, our TEM observations have enabled the unambiguous identification of twin domains in tetragonal CH₃NH₃PbI₃, ranging from around 100–300 nm wide, with twin boundaries parallel to {112}ₜ. As seen in the model of Fig. 3, the very small differences in the spacing of the {110}ₜ and {002}ₜ planes underpin the twin formation.

The twin geometry observed in the electron diffraction and imaging data presented here is clearly different to all of the various conflicting twin models proposed from X-ray data analysis[7,8,17]. Stoumpos et al.[7] and Fang et al.[8] both found that they could improve the fit to single-crystal X-ray diffraction data (taken at 293 K and 200 K, respectively) if they included 'pseudo-merohedral' twins in their model structure. However,

they used different twin models to improve their refinements. Stoumpos et al.[7] included twinning in their model via a 180° rotation about the [010]ₜ axis in the I4cm space group, whereas Fang et al.[8] included twinning via 120° rotations around the [201]ₜ zone axis in the I4/mcm space group, to form three twin domains at 120°. Hermes et al.[17] proposed a third model based on polycrystalline thin film data with (110)ₜ domain boundaries.

The determination of twin geometry from single-crystal electron diffraction patterns can be made by direct inspection and does not require numerical refinement of test models. Furthermore, the geometry can be correlated with the twin boundary contrast observed in the images. The twin domains observed here are not merohedric and involve a mirror relationship about the {112}ₜ plane. In particular, no twin boundaries were observed at 120° to each other in the images. This knowledge of the twin geometry will enable much better refinements of X-ray data, improving the refinement of atomic positions to provide superior insight into the intrinsic atomic structure of the unit cell.

It is important to note that, even with the low-dose imaging condition used here, the observed twin-domain contrast in images (Fig. 1d) and associated 'split' spots in SAED patterns (Fig. 1e,f) disappear very quickly under electron beam irradiation, making these features extremely easy to miss. Furthermore, we found this damage to be irreversible, even after in situ thermal annealing. This reflects the annihilation of the twinning structure associated with subtle compositional changes due to electron irradiation (we will describe this in depth in a paper currently in preparation), although the overall CH₃NH₃PbI₃ grain morphology and crystallinity remain intact (Fig. 1d,e). The fragility of the twins under the electron beam, as illustrated in Fig. 1, may be a reason why this twinning phenomenon has not previously been identified via TEM. It is for the same reason that we have not obtained atomic resolution TEM images of the twin boundary structure.

Note that the 100 index (or a = b parameter) in the cubic notation corresponds to the 110/002 indices in the tetragonal notation and similarly the {101}ᶜ plane in the cubic structure corresponds to the {112}ₜ plane in the tetragonal structure. In the model in Fig. 3a, differences in the spacing of $d_{110}$ and $d_{002}$ in the tetragonal structure underpin the twinning and formation of the {112}ₜ twin plane. The absence of this spacing difference in the cubic structure means that twin domains are not expected to form in this structure.

**Effects of phase transition on twin domains.** It has previously been speculated that twinning in CH₃NH₃PbI₃ forms during the cubic-to-tetragonal transition[7,12]. We investigated this claim about the origin of the twinning structures by heating the CH₃NH₃PbI₃ film inside the electron microscope and carried out an in situ observation at nominal 70 °C, leaving the specimen to heat up for 10 min before illuminating. This temperature was chosen to be sufficiently high to ensure a definite transformation into the cubic phase (the cubic-to-tetragonal transition temperature is at around 57 °C (ref. 7)) but not so high as to induce thermal degradation of CH₃NH₃PbI₃. At room temperature, the striped twin domains observed in the tetragonal phase were clearly visible (Fig. 6a), but disappeared upon heating to nominal 70 °C, transforming into a uniform contrast throughout all of the grains (Fig. 6b). To exclude the possibility of beam damage causing this contrast change, we cooled the film down to room temperature inside the microscope, and re-imaged the same area (slightly drifted from the original position after heating/cooling). After cooling, the striped domains

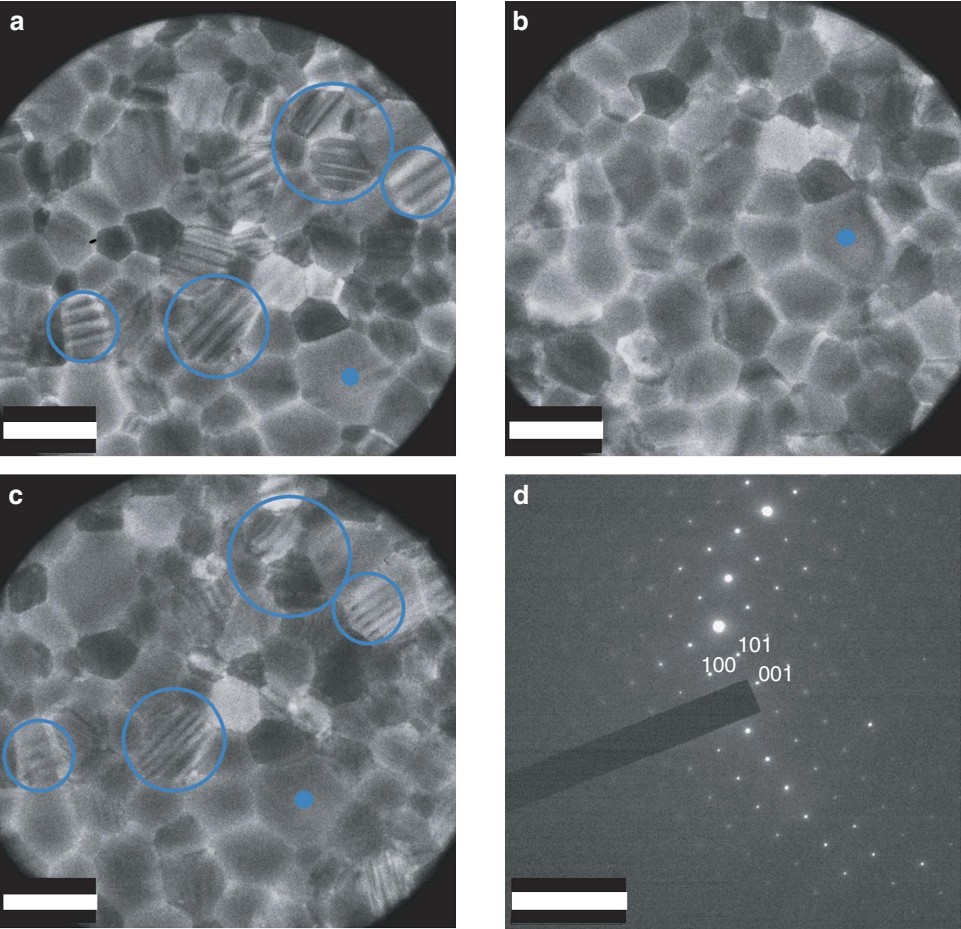

**Figure 6 | Effects of phase transition on twin domains in the CH$_3$NH$_3$PbI$_3$ thin film.** (**a–c**) Bright-field TEM images of the same area at the same orientation in a CH$_3$NH$_3$PbI$_3$ thin film (**a**) at room temperature, (**b**) heated to nominal 70 °C for 10 min and (**c**) cooled down to room temperature again. The film was only exposed briefly to record the images and the beam was turned off during the temperature ramping. The same grain in these images is marked with a blue dot as a reference for comparison. All of the striped twin domains disappear upon heating beyond the tetragonal-to-cubic transition temperature. Some twin domains reappear when the film is cooled to room temperature again, with some domains showing the same striped pattern as before heating (blue circles). (**d**) SAED pattern from a grain at nominal 70 °C indexed in the cubic phase and oriented near to the $<010>_c$ zone axis (equivalent to $<110>_t$ in the tetragonal phase), showing no diffraction spot 'splitting' and thus no twinning. Scale bar, 1 µm (**a–c**); 5 nm$^{-1}$ (**d**).

reappeared with the same domain orientation in some of the grains (blue circles in Fig. 6c). (In some grains, the stripe contrast is weaker, most likely due to beam damage.) The diffraction pattern obtained at nominal 70 °C shows a single-crystal diffraction pattern, without any 'splitting' of diffraction spots (Fig. 6d). These observations prove unambiguously that the twin domains form during the cubic-to-tetragonal transition in CH$_3$NH$_3$PbI$_3$, and their formation is reversible (possibly as a mechanism to release internal strain due to the slight difference between lattice parameters that occurs across the cubic-to-tetragonal transition[23], as suggested by Hermes *et al.*[17]). It is important to note that despite there being several possible ways to form twinning (such as mirroring across (112)$_t$ or (11$\bar{2}$)$_t$), most of the striped domains appear to keep the same orientation before and after heating, as highlighted with blue circles in Fig. 6a,c). Such a 'memory' effect may indicate the presence of certain constraints such as strain at grain boundaries[25], which determines the orientation of the twin domains. A similar twin memory effect was observed in ferroelastic materials due to enhanced defect density at the twin boundaries[26]. The defects can be relatively slow to migrate when heated above the tetragonal-to-cubic phase transition. Their presence can then provide an energy-efficient site for the reformation of the twin boundaries at

the same location when the material is cooled back into the tetragonal phase, as has been observed in a number of materials[26]. In CH$_3$NH$_3$PbI$_3$, the memory effect may be controlled by extended defects, such as the grain boundaries, and/or point defects, such as occasional methylammonium vacancies, for example.

In practical solar cell fabrications, CH$_3$NH$_3$PbI$_3$ is usually crystallized above the phase transition temperature of 57 °C and cooled down to room temperature before study or use[27,28]. It therefore inevitably undergoes a phase transition from a high-symmetry cubic phase to a low-symmetry tetragonal phase, with associated distortion of the unit cell and possible generation of long-range strain[29]. Due to the possible operation of PSC across a wide range of temperatures, including the tetragonal to cubic phase transformation temperature, any performance variation caused by stress related to twin formation and disappearance needs further study.

**Discussion**

Further investigation is required to determine the relevance of the twin domains to device performance. There are several points to consider in this respect.

The impact of twin boundaries on charge separation, transport and recombination[30–32]. This may be negative or positive, depending on the twin boundary orientation with respect to the device interface and on the detailed atomic structure at the boundaries, such as whether they contain vacancies or other defects[30–32]. These effects may be amplified or even controlled if the boundaries are decorated with defects (as might be suggested by the observed memory effect).

The width of the domains relative to the device thickness. When these are comparable, it can influence the degree and distribution of strain and hence the electronic band structure[33].

The behaviour of twin boundaries close to the tetragonal to cubic phase transition. This transition lies within the operating temperature range of these solar cell devices and the stability of the twin domains and/or any defects that may lie at their boundaries, may be relevant to device stability.

The existence of twin domains is not evidence for ferroelectricity. However, twin domains are associated with ferroelectric domains in some other perovskites[34], such as $BaTiO_3$ (ref. 24), raising the need for further study.

The above effects need to be understood, so they can be controlled and optimized. For example, there may be potential to improve device performance by tailoring the orientation of twin boundaries relative to the device interface or tuning the twin boundary width relative to device thickness to optimize strain.

The unequivocal identification of the presence, dimensions and orientation of these twin domains shown here will enable superior refinement of crystal structure from X-ray diffraction data, providing the structural information necessary to understand and predict critical properties and to further improve the performance of PSC.

Using TEM, we have provided direct and unequivocal evidence for the existence and crystallography of twin domains in tetragonal methylammonium lead tri-iodide ($CH_3NH_3PbI_3$) thin films used for solar cell applications. The twin domains range from around 100 to 300 nm in width and have a twin boundary parallel to $\{112\}_t$. Importantly, the absence/presence of the twin domains is reversible when cycling through the cubic/tetragonal phase transition, even the scale and orientation of the twin domains is largely 'memorized'. These twins have eluded observation so far, possibly due to their very fragile nature under the electron beam, as well as the inherent instability of the material itself. Given the scale of these domains is comparable to the thickness of typical methylammonium lead iodide perovskite layers used in solar cells, and given the twinning transition temperature lies within the operational temperature range of solar cells, these twin domains are likely to play an important role in the functional performance of PSC. Further study on the effects of twinning boundaries on the free-carrier transportation and recombination is needed to guide improvements of PSC in the future.

## Methods

**TEM specimen preparation.** The $CH_3NH_3PbI_3$ organic − inorganic perovskite structure is fragile and degrades readily with exposure to moisture[35]. Care must therefore be taken to use a TEM specimen method that delivers a pristine, undamaged structure. Popular methods such as focused ion beam (FIB) milling[36,37], argon milling or tripod polishing can cause significant damage to the crystal structure. To avoid this, a polycrystalline $CH_3NH_3PbI_3$ thin film was deposited directly onto a thin carbon-coated TEM grid by a gas-assisted rapid quenching method[27].

Precursor solutions were prepared by combining 99.9% pure lead iodide ($PbI_2$, Sigma-Aldrich), and methylammonium iodide ($CH_3NH_3I$, MAI, synthesized in-house) stoichiometrically in dimethylformamide, obtaining a 45 wt.% solution. A $CH_3NH_3PbI_3$ film was then deposited by spin coating the 45 wt.% $CH_3NH_3PbI_3$ precursor solution at 6,500 r.p.m. for 30 s, using a gas-assisted spin coating method, and then annealed at 100 °C for 10 min[27]. The film thickness was estimated from

a FIB cross-section of a similarly prepared TEM specimen and varies over the grid from around 100 to 300 nm.

**TEM characterization.** The TEM specimens were transferred to the TEM (a JEOL 2100F FEG-TEM with Gatan Ultrascan camera) in a dry atmosphere. To minimize possible electron beam-induced artefacts, we used a low-dose TEM imaging condition with an electron dose rate of around $1\,e\,Å^{-2}\,s^{-1}$. We employed a high-contrast objective aperture to increase the contrast of the twin domains. All the TEM images and diffraction patterns were recorded from previously unexposed regions of the sample, except on those occasions identified below where an image and diffraction pattern were specifically taken from the same area. In particular, the crystal grains were examined in an 'as-found' orientation, without any attempt at crystal alignment that would have incurred further electron dose. This approach had the added benefit that a very large number of different crystal grains and crystal orientations could be examined, ensuring good observational statistics and that all twin plane geometries present in the specimen will be detected.

In this study, two different approaches were used to examine the twin domains: first, direct and 'instant' recording in diffraction space (or image space) of an essentially pristine, as found, previously unexposed region. Second, an image of the domain contrast was taken and subsequently a diffraction pattern from a specific region in that image was taken, to correlate the domain contrast with the crystallographic information in the diffraction pattern. The first method allows for crystallographic information to be obtained from practically undamaged material, whereas the second approach necessarily incurs some additional electron dose, due to the time involved in switching from image mode to diffraction mode (adding about 30 s of weak (around $1\,e\,Å^{-2}\,s^{-1}$) electron exposure relative to the first approach). However, it enables a correlation of the striped image contrast with the corresponding diffraction pattern from that region, as shown in Fig. 4.

**SEM characterization.** For imaging the surface morphology of the film, a $CH_3NH_3PbI_3$-coated TEM grid was attached to carbon tape and placed in an FEI Helios Nanolab600 Dual Beam FIB-SEM. The images were recorded using 2 kV acceleration voltage and 13 pA probe current. The dwell time for the recording was 10 μs per probe pixel.

**Data availability.** The data that support the findings of this study are available from the corresponding authors upon request.

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

## Acknowledgements

This work was financially supported by the Australian Government through the Australian Renewable Energy Agency and the Australian Centre for Advanced Photovoltaics. Responsibility for the views, information or advice expressed herein is not accepted by the Australian Government. The authors acknowledge use of facilities within the Monash Centre for Electron Microscopy. M.R. and W.L. is grateful to Dr Laure Bourgeois for expert advice on the operation of the JEOL 2100F. This work was performed in part at the Melbourne Centre for Nanofabrication in the Victorian Node of the Australian National Fabrication Facility. The authors thank Professor Fuzhi Huang from Wuhan University of Technology for the valuable discussion.

## Author contributions

M.U.R., W.L., Y.Z., J.E. and Y.-B.C. conceived and designed the experiment. M.U.R. and W.L. carried out sample preparation. M.U.R., W.L., Y.Z. and J.E. did electron microscopy and data analysis. All authors contributed to the discussion of the results and to the writing of the manuscript.

## Additional information

**Competing financial interests:** The authors declare no competing financial interests.

