## [Peer Review File · Nature Communications]

Reviewers' comments:

Reviewer #1 (Remarks to the Author):

The manuscript by Rothmann et al. reports on the twin domain structure in the tetragonal (room temperature) phase of the well-known perovskite methylammonium lead iodide. Although twinning has previously been observed by X-ray diffraction in this phase (Refs. 7 and 8 of the manuscript), the precise nature of the twinning has remained unclear. This is despite the fact that twinning (especially if this involves ferroelectric domains) might have an important influence on the properties of the material in photovoltaic devices. Taking into account the huge number of publications on this material in recent years, the current study is overdue and of potentially great importance. In my opinion this study could have a large impact on the field of hybrid perovskite photovoltaics. However, I feel that more insight into the nature of the domains could and should be provided before publication in Nature Communications can be considered.

1. I am not convinced that the twinning only occurs via 112 mirror planes. In Figure S3 of the supplementary information domains are observed that appear almost perpendicular to each other within the same grain. Can the authors comment on the twin planes in this grain and how these 90 degree rotated domains are related to each other? Furthermore, close inspection suggests that the "horizontal" domains in this grain taper from right to left in the figure (they are wider on the right-hand side than on the left-hand side). Therefore, the "upper" and "lower" domain walls are not exactly parallel. Can all the twin planes in this picture be identified? What is the zone axis here?

2. Is spot splitting observed for zone axes other than [1-10]? For example, in the caption to figure 3 the authors cite a lack of spot splitting in the [010] zone axis diffraction pattern as evidence for the lack of twinning above 70 °C. This implies that they observed spot splitting below the phase transition in this zone axis. Also, how can the authors be sure that the twin planes in the [1-10] projection of Figure 1 are precisely perpendicular to the viewing direction? Analysis of more zone axes is necessary in my opinion.

3. Twinning was previously observed and characterised by the XRD studies of Stoumpos et al. (Ref. 7) and Fang et al. (Ref. 8). Is the current twinning model consistent with these previous reports or not? A discussion of this aspect would be very useful.

Reviewer #2 (Remarks to the Author):

Review of "Direct observation of intrinsic twin domains in tetragonal CH₃NH₃PbI₃" by Mathias Uller Rothmann, Wei Li, Ye Zhu, Udo Bach, Leone Spiccia, Joanne Etheridge, and Yi-Bing Cheng

In their manuscript "Direct observation of intrinsic twin domains in tetragonal CH₃NH₃PbI₃" the authors describe the observation of crystallographic twin domains in tetragonal MAPbI₃ films by low-dose TEM and selected area electron diffraction. In the TEM images they found stripes with a width of 100-300 nm of alternating dark and bright contrast. The authors propose that twin domains of different crystallographic orientations induce the stripes with alternating intensities. An influence of the surface morphology was excluded by secondary electron imaging by SEM.

In diffraction patterns obtained via SAED measurements the authors observed split spots, reflections mirrored across a twin axis, which also indicate twinning. Especially, the reflections of the 220/004 family showed an increasing separation between the split spots as the distance to the origin increased. Based on their SEAD results the authors proposed a crystallographic model for the orientation in these twin domains with the mirror plane of the twin domains (twin boundaries) parallel to the (112) crystal plane. Due to the tetragonal MAPbI₃ adjacent twin domains exhibit a 89° angle between their (001) direction. The authors compare the assigned twin orientations to the 90° a-a domains on BTO.

Additionally, the authors found that upon heating above the cubic-tetragonal phase transition the domains disappeared and reappeared upon cooling at the same positions as before, concluding a memorizing effect of the material.

The authors of this manuscript conducted a well-researched study on their observation of crystallographic twins by TEM, discussed their results thoroughly and conclusively, and including the electron beam induced damage of MAPbI₃. The manuscript is very well written and easy to understand. While twin domains in MAPbI₃ have been previously observed by PFM (Hermes et al.) this TEM study includes measurements above the cubic-tetragonal phase transition and the authors proposed a new model for the orientation of the crystallographic twins. While this manuscript is most certainly of interest for researchers in the field of perovskite solar cells, it is however lacking in novelty. Further indications as to how the observations relate to the extraordinary performance of perovskite in solar cells would certainly increase the impact of this manuscript.

Minor remarks:

Abstract lines 33-35: "...and their fundamental implications for important properties such as charge separation, transport and recombination..." as this was not subject of their study the authors should not mention these points in the abstracts but rather as an outlook in the conclusion.

Line 91: "...low-dose, rapid acquisition..." some numbers on the time scales of those measurements might help the readers here.

Lines 92 & 93: "... the size of the twin domains is comparable to the thickness of the perovskite layer in a photovoltaic device." Here, it would be helpful to already mention the dimensions of the domains and the film thickness.

Figure 1 is captioned with Figure 2

Questions:

For the heating experiment: Is the weaker contrast after the cooling down caused by beam-induced sample damage?

What is the possible reason for the vanishing of the twin domains with longer electron beam exposure if the crystals stay intact? Do the domains recover after irradiation, i.e. is the damage reversible?

I did not quite understand which crystal plane equals the surface on which those stripes are apparent. Hermes et al. concluded from their 2D XRD data that they observed the stripes in PFM on the (110) crystal plane. Here, it is the (11 $\bar{0}$) plane?

It would also be helpful if the authors add an explanation as to why only some of the grains in Figure 1 exhibited the stripes, while other did not.

Reviewer #3 (Remarks to the Author):

This well-written manuscript reports the direct observation and confirmation of twins in polycrystalline CH₃NH₃PbI₃ hybrid perovskites using transmission electron microscopy (TEM). The authors also show the disappearance of the twins when the TEM specimen is heated above the tetragonal-to-cubic transformation (~60 C), and twin formation upon cooling, with some twinning 'memory.' The authors use specimen that is directly deposited on the TEM grid to avoid specimen-preparation artifacts, and they use a low-dose imaging technique to reduce electron-beam damage to the specimen during TEM observation. The TEM characterization and the crystallographic analysis are performed competently -- very convincing results. This revelation and confirmation of twinning in CH₃NH₃PbI₃, although not entirely surprising, will be useful to the hybrid-perovskites community.

There are few issues the authors should consider addressing.

1. The statement on lines 56-57 "...not least because a polar space group is a prerequisite for the presence of ferroelectricity..." is not entirely correct in hybrid perovskites. This rule assumes spherical atoms in ABX₃ compounds. But in CH₃NH₃PbI₃, the A site is occupied by a non-spherical molecular cation, and, thus, even a cubic arrangement of PbI₆ octahedra can result in ferroelectricity where non-random orientation of CH₃NH₃ cation can break the symmetry (see e.g. Butler, et al. in Environ. Ener. Sci. (2014) doi: 10.1039/C4EE03523B).
2. The speculation that strain is somehow responsible for the stochastic nature of the twinning appears reasonable, but more substantive explanation/quantification should be provided.
3. Also, there is no substantive explanation given for the appearance and disappearance of the twinning through heat-treatment, and the 'memory.' The latter is to be expected, considering that the tetragonal-to-cubic transformation is non-reconstructive. However, it should be noted that cubic materials twin readily (e.g. Cu, Au, etc.). Thus, it is likely that there are other effects involved.
4. Contrary to the authors' suggestion, the size of the twins matching the film thickness actually makes twinning less important. Small number, or no twin boundaries, within the thickness of the film would have minimal effect on carriers recombination, scattering, etc. (As a corollary, twin boundaries would be very important in a 'nano-twinned' CH₃NH₃PbI₃.)
5. Twins have coherent, low-energy boundaries, and hence their importance in the context of solar cells is going to be significantly less important than the regular (high-angle) grain-boundaries between random grains that proliferate the microstructures of hybrid perovskites. Thus, the overall importance of twin boundaries in the context of solar cells is debatable, vis-a-vis regular grain boundaries that are expected to dominate any microstructural effects.
6. Some experimental details are missing, such as the TEM accelerating voltage used. Also, although the nominal thickness of the thin film specimen is reported to be 300 nm, it is not clear how electron-transparency is obtained in such a thick film. Lettering in Figures 2b and 2e is too small.

Black indicates reviewer comments, blue indicates author response, red indicates author changes to manuscript. We have also marked up the changes in the revised manuscript.

Reviewer #1:

The manuscript by Rothmann et al. reports on the twin domain structure in the tetragonal (room temperature) phase of the well-known perovskite methylammonium lead iodide. Although twinning has previously been observed by X-ray diffraction in this phase (Refs. 7 and 8 of the manuscript), the precise nature of the twinning has remained unclear. This is despite the fact that twinning (especially if this involves ferroelectric domains) might have an important influence on the properties of the material in photovoltaic devices. Taking into account the huge number of publications on this material in recent years, the current study is overdue and of potentially great importance. In my opinion this study could have a large impact on the field of hybrid perovskite photovoltaics. However, I feel that more insight into the nature of the domains could and should be provided before publication in Nature Communications can be considered.

Reviewer 1, comment 1: I am not convinced that the twinning only occurs via 112 mirror planes.

We thank the referee to alerting us to a lack of clarity in the text. We have modified the text to address this, as follows:

Page 5, Line 120 – Inserted after “...Figure 1(c)”:1

“A comprehensive survey of reciprocal space was undertaken (see TEM characterisation) and all diffraction patterns were consistent with this twinning geometry. No other intrinsic twin geometries were observed.”

Page 14, Line 337 – Inserted after “...dose.”

“This approach had the added benefit that a very large number of different crystal grains and crystal orientations could be examined, ensuring good observational statistics and that all twin plane geometries present in this specimen will be detected.”

Reviewer 1, comment 1 continued: In Figure S3 of the supplementary information domains are observed that appear almost perpendicular to each other within the same grain. Can the authors comment on the twin planes in this grain and how these 90 degree rotated domains are related to each other?

Furthermore, close inspection suggests that the "horizontal" domains in this grain taper from right to left in the figure (they are wider on the right-hand side than on the left-hand side). Therefore, the “upper” and “lower” domain walls are not exactly parallel. Can all the twin planes in this picture be identified? What is the zone axis here?

Page 7, Line 159 – Inserted after “...including”:

“orthogonal domain boundaries and associated needle shapes typical of ferroelastic crystals²³”

and delete “. a 90°-direction-change of the domain boundaries”

Supp Info – Figure S3 – New caption

“Grain from a $\text{CH}_3\text{NH}_3\text{PbI}_3$ thin film showing two sets of twin domains orthogonal to each other representing different members of the symmetry-equivalent family of planes $\{112\}_t$ [e.g. $(112)_t$ and $(11\bar{2})_t$ which are oriented $\sim 89.5^\circ$ to each other]. (To avoid the additional electron dose that would be incurred through tilting, the grain was imaged ‘as found’, tilted off the $\langle 110 \rangle_t$ zone axis.) The ‘horizontal’ domains taper as they approach the intersection

with their orthogonal analogue: this is behaviour typical of twin domains in a wide range of ferroelastic crystals¹. These needle-like shapes form to minimise strain arising from the interaction with an orthogonal domain (or other local ‘defect’)². The resultant stress provides sufficient energy to enable a local deviation from the primary twin orientation³”.

Reviewer 1, comment 2: Is spot splitting observed for zone axes other than [1-10]?

This question is addressed in our response to comment (1) above, which we repeat here for ease of reading:

Page 5, Line 120– Inserted after “...Figure 1(c)”:

“A comprehensive survey of reciprocal space was undertaken (see TEM characterisation) and all diffraction patterns were consistent with this twinning geometry. No other intrinsic twin geometries were observed.”

Page 14, Line 337 – Inserted after “...dose.”

“This approach had the added benefit that a very large number of different crystal grains and crystal orientations could be examined, ensuring good observational statistics and that all twin plane geometries present in this specimen will be detected.”

Reviewer 1, comment 2 continued: For example, in the caption to figure 3 the authors cite a lack of spot splitting in the [010] zone axis diffraction pattern as evidence for the lack of twinning above 70 °C. This implies that they observed spot splitting below the phase transition in this zone axis.

The diffraction pattern in Figure 3 was taken at 70° C (above the tetragonal to cubic phase transition) and hence the diffraction pattern was indexed in the cubic phase, not the tetragonal phase (we have updated Figure 3 (d) to include a diffraction pattern without interference from a neighbouring grain). The $\langle 010 \rangle_c$ direction in the cubic phase is equivalent to the $\langle 110 \rangle_t$ direction in the tetragonal phase. “Spot splitting” was indeed observed below the phase transition in this zone axis, i.e. the $\langle 110 \rangle_t$ zone axis.

We appreciate this was poorly expressed and have added the following text to clarify this point:

Page 4, Line 78 – Text modified and added after “...angle to the “

$\{110\}_t$ surface (throughout this manuscript, the subscripts “t” and “c” denote indexing in the tetragonal phase and cubic phase, respectively.)

Figure 3 Caption – Page 23, Line 539 (Page 20, 509) – Inserted text after “...grain at 70 C”:

“ , indexed in the cubic phase and oriented near to the $\langle 010 \rangle_c$ zone axis (equivalent to $\langle 110 \rangle_t$ in the tetragonal phase),”

Reviewer 1, comment 2 continued: Also, how can the authors be sure that the twin planes in the [1-10] projection of Figure 1 are precisely perpendicular to the viewing direction?

As described above, the diffraction patterns taken from many different *individual* grains in many different orientations provide unequivocal evidence for the twin plane geometry.

In the images in Figure 1, not all grains will be oriented parallel to $[1\bar{1}0]_t$ in the field of view. To clarify this point, the following text has been added:

Page 6, Line 130 – text added at the end of this line:

“It is noted that some of the grains in Figure 1 (a) will be oriented so that the electron beam is at a significant angle to the twin boundary. In these projections, little contrast will be visible. In particular, if the angle is 90° , no contrast will be visible.”

Reviewer 1, comment 2 continued: Analysis of more zone axes is necessary in my opinion.

As explained above, we did in fact do a comprehensive survey of reciprocal space but did not make this clear in the original manuscript. We trust the additional text (described above) has now clarified this point.

Reviewer 1, comment 3: Twinning was previously observed and characterised by the XRD studies of Stoumpos et al. (Ref. 7) and Fang et al. (Ref. 8). Is the current twinning model consistent with these previous reports or not? A discussion of this aspect would be very useful.

We have expanded on our discussion of previous XRD studies with the following additional text:

Page 8, Line 178 – the following additional paragraphs have been inserted:

The twin geometry observed in the electron diffraction and imaging data presented here is clearly different to all of the various conflicting twin models proposed from x-ray data analysis^{7,8,17}. Stoumpos et al.⁷ and Fang et al.⁸ both found that they could improve the fit to single-crystal X-ray diffraction data (taken at 293 K and 200 K, respectively) if they included “pseudo-merohedral” twins in their model structure. However, they used different twin models to improve their refinements. Stoumpos et al.⁷ included twinning in their model via a 180° rotation about the $[010]_t$ axis in the $I4cm$ space group, whereas Fang et al. included twinning via 120° rotations around the $[201]_t$ zone axis in the $I4/mcm$ space group, to form three twin domains at 120°. Hermes et al.¹⁷ proposed a third model based on polycrystalline thin film data with $(110)_t$ domain boundaries.

The determination of twin geometry from single crystal electron diffraction patterns can be made by direct inspection and does not require numerical refinement of test models. Furthermore, the geometry can be correlated with the twin boundary contrast observed in the images. The twin domains observed here are not merohedric and involve a mirror relationship about the $\{112\}_t$ plane. In particular, no twin boundaries were observed at 120° to each other in the images. This knowledge of the twin geometry will enable much better refinements of X-ray data, improving the refinement of atomic positions to provide superior insight into the intrinsic atomic structure of the unit cell.

Reviewer #2:

In their manuscript “Direct observation of intrinsic twin domains in tetragonal $CH_3NH_3PbI_3$ ” the authors describe the observation of crystallographic twin domains in tetragonal $MAPbI_3$ films by low-dose TEM and selected area electron diffraction. In the TEM images they found stripes with a width of 100-300 nm of alternating dark and bright contrast. The authors propose that twin domains of different crystallographic orientations induce the stripes with alternating intensities. An influence of the surface

morphology was excluded by secondary electron imaging by SEM. In diffraction patterns obtained via SAED measurements the authors observed split spots, reflections mirrored across a twin axis, which also indicate twinning. Especially, the reflections of the 220/004 family showed an increasing separation between the split spots as the distance to the origin increased. Based on their SEAD results the authors proposed a crystallographic model for the orientation in these twin domains with the mirror plane of the twin domains (twin boundaries) parallel to the (112) crystal plane. Due to the tetragonal MAPbI₃ adjacent twin domains exhibit a 89° angle between their (001) direction. The authors compare the assigned twin orientations to the 90° a-a domains on BTO.

Additionally, the authors found that upon heating above the cubic-tetragonal phase transition the domains disappeared and reappeared upon cooling at the same positions as before, concluding a memorizing effect of the material.

The authors of this manuscript conducted a well-researched study on their observation of crystallographic twins by TEM, discussed their results thoroughly and conclusive, and including the electron beam induced damage of MAPbI₃. The manuscript is very well written and easy to understand. While twin domains in MAPbI₃ have been previously observed by PFM (Hermes et al.) this TEM study includes measurements above the cubic-tetragonal phase transition and the authors proposed a new model for the orientation of the crystallographic twins.

While this manuscript is most certainly of interest for researchers in the field of perovskite solar cells, it is however lacking in novelty. Further indications as to how the observations relate to the extraordinary performance of perovskite in solar cells would certainly increase the impact of this manuscript.

As far as we are aware, this manuscript is the first to provide direct and unequivocal identification of the twin domains and their geometry in the hybrid perovskite. The excellent paper of Hermes et al (ref 17) observed periodic domains in the piezoresponse and endeavoured to correlate this with polycrystalline x-ray data to identify the polarization

direction. However, it is notoriously difficult to determine twin geometry from polycrystalline X-ray data and, indeed, X-ray data in general, as this requires numerical refinement of hypothetical models with the inherent possibility of non-unique solutions. It is therefore maybe not surprising that the three X-ray studies have each proposed different models for the twin geometry.

Reviewer 2 continued: Abstract lines 33-35: “...and their fundamental implications for important properties such as charge separation, transport and recombination...” as this was not subject of their study the authors should not mention these points in the abstracts but rather as an outlook in the conclusion.

Page 2, Line 33-35: We have deleted

“...and their fundamental implications for important properties such as charge separation, transport and recombination...” .

Reviewer 2 continued: Line 91: “...low-dose, rapid acquisition...” some numbers on the time scales of those measurements might help the readers here.

Page 4, Line 91: Inserted after “...low dose,”:

(~1 e/(Å² s))

Reviewer 2 continued: Lines 92 & 93: ...” the size of the twin domains is comparable to the thickness of the perovskite layer in a photovoltaic device.” Here, it would be helpful to already mention the dimensions of the domains and the film thickness.

Page 4, Line 93: the text in red has been inserted:

...” the size of the twin domains (~100-300 nm) is comparable to the thickness of the perovskite layer (~300 nm) in a photovoltaic device.”

Figure 1 is captioned with Figure 2

We thank the referee for picking this up. This has been corrected.

Reviewer 2 continued: Question 1: For the heating experiment: Is the weaker contrast after the cooling down caused by beam-induced sample damage?

Page 10, Line 232: Text inserted after "...Figure 3 (c).":

"In some grains, the stripe contrast is weaker, most likely due to beam damage."

Reviewer 2 continued: Question 2: What is the possible reason for the vanishing of the twin domains with longer electron beam exposure if the crystals stay intact? Do the domains recover after irradiation, i.e. is the damage reversible?

Page 8, Line 203: Text inserted after "...twinning structure"

"associated with subtle compositional changes²⁵"

Page 9, Line 201: Text inserted after "...easy to miss."

"Furthermore, we found this damage to be irreversible, even after *in-situ* thermal annealing."

Reviewer 2 continued: Question 3: I did not quite understand which crystal plane equals the surface on which those stripes are apparent. Hermes et al. concluded from their 2D XRD data that they observed the stripes in PFM on the (110) crystal plane. Here, it is the (11 $\bar{0}$) plane?

We had already noted in the original text that the mirror plane (that is, the domain boundary) is parallel to $\{112\}_t$ (e.g. in lines 27, 133, 185, 209, 210, 240 etc. in the original document). We have therefore not added any additional text to address this question.

It would also be helpful if the authors add an explanation as to why only some of the grains in Figure 1 exhibited the stripes, while other did not.

Page 6, Line 130– text added at the end of this line:

"It is noted that some of the grains in Figure 1 (a) will be oriented so that the electron beam is at a significant angle to the twin boundary. In these projections, little contrast will be visible. In particular, if the angle is 90°, no contrast will be visible."

Reviewer #3:

This well-written manuscript reports the direct observation and confirmation of twins in polycrystalline CH₃NH₃PbI₃ hybrid perovskites using transmission electron microscopy (TEM). The authors also show the disappearance of the twins when the TEM specimen is heated above the tetragonal-to-cubic transformation (~60 C), and twin formation upon cooling, with some twinning ‘memory.’ The authors use specimen that is directly deposited on the TEM grid to avoid specimen-preparation artifacts, and they use a low-dose imaging technique to reduce electron-beam damage to the specimen during TEM observation. The TEM characterization and the crystallographic analysis are performed competently -- very convincing results. This revelation and confirmation of twinning in CH₃NH₃PbI₃, although not entirely surprising, will be useful to the hybrid-perovskites community.

There are few issues the authors should consider addressing.

Reviewer 3, Question 1: The statement on lines 56-57 “...not least because a polar space group is a prerequisite for the presence of ferroelectricity...” is not entirely correct in hybrid perovskites. This rule assumes spherical atoms in ABX₃ compounds. But in CH₃NH₃PbI₃, the A site is occupied by a non-spherical molecular cation, and, thus, even a cubic arrangement of PbI₆ octahedra can result in ferroelectricity where non-random orientation of CH₃NH₃ cation can break the symmetry (see e.g. Butler, et al. in Environ. Ener. Sci. (2014) doi: 10.1039/C4EE03523B).

We thank the reviewer for bringing our attention to this interesting paper.

In its discussion of this scenario, Butler et al note that the overall crystal symmetry is reduced: the BX₃ octahedra may be located at cubic lattice sites but the non-random orientation of the A cation breaks the symmetry of the unit cell as a whole. We believe this is consistent with what is written in lines 56-57. However, if the reviewer would prefer, we can modify this text as follows:

Page 3, Lines 56-57:

“This is an important question to resolve. Crystal structure controls properties, including ferroelectricity, which has been proposed to possibly...”

Reviewer 3, Question 2. The speculation that strain is somehow responsible for the stochastic nature of the twinning appears reasonable, but more substantive explanation/quantification should be provided.

Page 10, Line 237: The following text has been added after “...internal strain,”

“due to the slight difference between lattice parameters that occurs across the cubic to tetragonal transition²³.”

Page 7, Line 159 – Inserted after “...including”:

“orthogonal domain boundaries and associated needle shapes typical of ferroelastic crystals²³ (see supplementary Figure S3) ...”

Supp Info – Figure S3 – New caption

“Grain from a $\text{CH}_3\text{NH}_3\text{PbI}_3$ thin film showing two sets of twin domains orthogonal to each other representing different members of the symmetry-equivalent family of planes $\{112\}_t$ [e.g. $(112)_t$ and $(11\bar{2})_t$ which are oriented $\sim 89.5^\circ$ to each other]. (To avoid the additional electron dose that would be incurred through tilting, the grain was imaged ‘as found’, tilted off the $\langle 110 \rangle_t$ zone axis.) The ‘horizontal’ domains taper as they approach the intersection with their orthogonal analogue: this is behaviour typical of twin domains in a wide range of ferroelastic crystals¹. These needle-like shapes form to minimise strain arising from the interaction with an orthogonal domain (or other local ‘defect’)². The resultant stress provides sufficient energy to enable a local deviation from the primary twin orientation³”.

Reviewer 3, Question 3. Also, there is no substantive explanation given for the appearance and disappearance of the twinning through heat-treatment, and the ‘memory.’ The latter is to be expected, considering that the tetragonal-to-cubic transformation is non-reconstructive. However, it should be noted that cubic materials twin readily (e.g. Cu, Au, etc.). Thus, it is likely that there are other effects involved.

To address this question, we have added the following text:

Page 10, Line 275: The following reference was inserted after “grain boundaries”:

“Xu, H. & Heaney, P. J. Memory effects of domain structures during displacive phase transitions; a high-temperature TEM study of quartz and anorthite. *Am. Mineral.* **82**, 99-108, (1997).”

Page 10, Line 245: At the end of the sentence, the following text was added:

“The defects can be relatively slow to migrate when heated above the tetragonal to cubic phase transition. Their presence can then provide an energy-efficient site for the reformation of the twin boundaries at the same location when the material is cooled back into the tetragonal phase, as has been observed in a number of materials²⁷. In $\text{CH}_3\text{NH}_3\text{PbI}_3$, the memory effect may be controlled by extended defects, such as the grain boundaries, and/or point defects, such as occasional methylammonium vacancies, for example.”

Reviewer 3, Question 4. Contrary to the authors’ suggestion, the size of the twins matching the film thickness actually makes twinning less important. Small number, or no twin boundaries, within the thickness of the film would have minimal effect on carriers recombination, scattering, etc. (As a corollary, twin boundaries would be very important in a ‘nano-twinned’ $\text{CH}_3\text{NH}_3\text{PbI}_3$.)

Reviewer 3, Question 5. Twins have coherent, low-energy boundaries, and hence their importance in the context of solar cells is going to be significantly less important than the regular (high-angle) grain-boundaries between random grains that proliferate the microstructures of hybrid perovskites. Thus, the overall importance of twin boundaries in the context of solar cells is debatable, vis-a-vis regular grain boundaries that are expected to dominate any microstructural effects.

We address Q4 and Q5 together and have endeavoured to clarify these points through the addition of the following text:

Page 11, Line 262 deleted.

Page 11, Line 262: The following text has been inserted:

Discussion

Further investigation is required to determine the relevance of the twin domains to device performance. There are several points to consider in this respect.

1. The impact of twin boundaries on charge separation, transport and recombination³¹⁻³³. This may be negative or positive, depending on the twin boundary orientation with respect to the device interface and on the detailed atomic structure at the boundaries, such as whether they contain vacancies or other defects³¹⁻³³. These effects may be amplified or even controlled if the boundaries are decorated with defects (as might be suggested by the observed memory effect).
2. The width of the domains relative to the device thickness. When these are comparable, it can influence the degree and distribution of strain and hence the electronic band structure³⁴.
3. The behaviour of twin boundaries close to the tetragonal to cubic phase transition. This transition lies within the operating temperature range of these solar cell devices and the stability of the twin domains and/or any defects that may lie at their boundaries, may be relevant to device stability.
4. The existence of twin domains is not evidence for ferroelectricity. However, twin domains are associated with ferroelectric domains in some other perovskites³¹, such as BaTiO₃²⁴, raising the need for further study.

The above effects need to be understood, so they can be controlled and optimised. For example, there may be potential to improve device performance by tailoring the orientation of twin boundaries relative to the device interface or tuning the twin boundary width relative to device width to optimise strain.

Reviewer 3, Question 6. Some experimental details are missing, such as the TEM accelerating voltage used. Also, although the nominal thickness of the thin film specimen is reported to be 300 nm, it is not clear how electron-transparency is obtained in such a thick film. Lettering in Figures 2b and 2e is too small.

We thank the referee for picking up these omissions. We have now added accelerating voltage, enlarged the lettering, and specified that the indexing is in the tetragonal phase.

Regarding the question of film thickness, the film is indeed thick for TEM but is sufficiently thin to enable the low resolution imaging and diffraction study undertaken here. We have added the following text:

Page 13, Line 321: delete from the start:

“~300 nm thick”

Page 13, Line 323: Insert at the end:

“The film thickness was estimated from a FIB cross-section of a similarly prepared TEM specimen and varies over the grid from ~100 nm to ~300 nm.

Reviewers' comments:

Reviewer #1 (Remarks to the Author):

The revisions made have substantially improved the manuscript and the authors have addressed all my previous concerns:

1. It is now clear that a comprehensive survey of reciprocal space was made and that only twinning via (112) mirror planes was observed. The 90 degree domains and tapering of contrast in Figure S3 are now explained in a satisfactory manner.

2. The authors have made it clear that spot splitting was indeed observed below 70 °C in the [010] zone axis diffraction pattern. It is also now explained that all grains were imaged "as found" and that the twin planes are sometimes tilted with respect to the viewing direction.

3. A detailed comparison with previous X-ray diffraction studies has been added. This was in my opinion necessary and I am satisfied with it.

In summary, the results presented are now much more convincing and might have a big impact on the field of hybrid perovskite photovoltaics. Therefore in my opinion the manuscript is now suitable for publication in Nature Communications.

One minor correction is needed: The caption to Figure 1 is labelled as Figure 2.

Reviewer #2 (Remarks to the Author):

The authors of the manuscript 'Direct observation of intrinsic twin domains in tetragonal CH₃NH₃PbI₃' have answered the questions and doubts of the reviewers thoroughly and improved their well-written TEM study accordingly.

I suggest adding some minor changes for further clarifications to the manuscript but overall recommend its publication in Nature Communications without further revision.

As a general question that could be discussed in the manuscript: Since the observation of twin domains is so sensitive to beam irradiation is there a possibility that the twin domains only exist in the first few surface layers of a perovskite grain, or is it certain that they penetrate through the whole grain?

For the heating experiments: specify if 70°C is the sample temperature or the temperature of the heating stage (sample temp. was probably lower in the case)

Supplementary Figure S1: Specify if this is the same film as used for the TEM measurements in Figure 1 in the main manuscript.

Including the minor revisions suggested here, I am confident that the study meets the requirements for a publication in Nature Communications and that the scientific community will benefit from its publication.

Reviewer #3 (Remarks to the Author):

The revisions are satisfactory, and the paper is acceptable as-is.

Following are our responses to the questions raised by two of the referees.

Reviewer #1

Q: One minor correction is needed: The caption to Figure 1 is labelled as Figure 2.

A: We have checked both the manuscript and the Supp Info and found the figure numbers in all figure captions are properly stated.

Reviewer #2

Q: As a general question that could be discussed in the manuscript: Since the observation of twin domains is so sensitive to beam irradiation is there a possibility that the twin domains only exist in the first few surface layers of a perovskite grain, or is it certain that they penetrate through the whole grain?

A: We believe the twinning observed in this work is a bulk effect, rather than a surface phenomenon. Under TEM the electron beam must fully transmit through the specimen. As a result the images collected from TEM reflect the bulk characteristics of the material. If the twinning bands only existed on the very top surface of an untwined grain, then an electron diffraction pattern of the untwined phase should be superimposed on that of the twinned domains. However we see no evidence for an untwined phase in the diffraction patterns of the twinned grains. We have clarified this point in the revised manuscript.

Q: For the heating experiments: specify if 70°C is the sample temperature or the temperature of the heating stage (sample temp. was probably lower in the case)

A: 70°C was the temperature of the TEM sample holder (copper). The specimen was on a copper grid (3 mm in diameter) mounted on the sample holder in a vacuum TEM chamber. We therefore think the actual specimen temperature was very close to 70°C if it was not exactly at the temperature after 10 minute heating. We have clarified 70°C to be a nominal temperature in the revised manuscript.

Q: Supplementary Figure S1: Specify if this is the same film as used for the TEM measurements in Figure 1 in the main manuscript.

A: It was a different film made in exactly the same way as that used for the TEM study in Figures 1 and 3. This has been clarified in the caption of Figure S1.

All the changes have been marked in the revised Manuscript and Supp Info for your consideration.